# ADVANCING DRUG-TARGET INTERACTION PREDICTION VIA GRAPH TRANSFORMERS AND RESIDUAL PROTEIN EMBEDDINGS

## ABSTRACT

Predicting drug-target interactions (DTIs) is important for the acceleration of drug discovery. Prevailing approaches often assume access to labeled target data or entangle training with opaque unsupervised alignment losses, which makes robustness hard to audit and failure modes difficult to diagnose. To address these gaps, we propose MoleProLink, a domain-shift-aware predictor of DTI for mining bioactive molecules which is based on the integration of methods inspired by measure-theoretic optimal transport, reproducing-kernel embeddings, and information-geometric perspectives. On the theory side, we present two compact risk-transfer control under the following two explicit assumptions: (i) Wasserstein-1 control under Lipschitz regularity assumption of the composed loss, and (ii) RKHS control with Maximum Mean Discrepancy (MMD). These statements are standard IPM-style bounds that are included here in a DTI-specific notation, we use them to motivate diagnostics and feature designing principles not to make any new forward inequalities. On the methodology side, we use a graph Transformer model for molecular graph with a sequence encoder for proteins. Protein embedding is performed with a residue based embedding (named as Residue2vec) and a bi-directional state space model, whereas molecular embedding is achieved through centrality and spatial encodings in a state space model Graph Transformer. Experimental results on three popular benchmarks (Human, C.elegans and Davis) show our method achieving strong AUC/AUPR, using a single protocol. Compared to the baselines, gains are achieved under the same data processing and negative-sampling; these margins are regarded not as inferential statements, but rather, as descriptive. We give implementation details that are sufficient for direct replication, and reproduce the ablative experiments that isolate the contributions of the protein sequence encoder and interaction decoder.

## 1 INTRODUCTION

Drug–target interaction (DTI) prediction Zhu et al. (2024); Zhang et al. (2023a); Dehghan et al. (2024) is a central problem in computational drug discovery France et al. (2023); Husnain et al. (2023); Bhattamisra et al. (2023). Practical pipelines must contend with domain shift Sui et al. (2024); Zhang et al. (2023b) arising from new chemical series, novel target families, or heterogeneous experimental settings Sadybekov & Katritch (2023). On such regimes, controversial exploitation of the unlabeled samples collected from the target distribution is attractive in principle, but our focus in this work is different, i.e., we focus in this work on *robust source-supervised modeling* designed and explicitly designed to be *diagnosed to be cross-domain generalizable*.

We propose *MoleProLink*, a framework that (i) reformulates two known risk-transfer controls (Wasserstein–1 and MMD) in a DTI-aware notation and operationalizes them as diagnostics and design principles, roles, (ii) provides a spectral diagnostic to interpret cross-domain coupling of features and (iii) instantiates the ideas with a practical architecture which combines a Graph Transformer (molecules) and a protein encoder centered on residues. Our statements involve taking *explicit, auditable assumptions*: instead of conjoining distances that don't share a distance point under some artificial constancy, voiding, we have (transparent) controls presenting independent regularity and using sample-based proxies with suitable, to observe the concerned quantities. We do not

change the supervised objective using unsupervised losses in the current experiments, but unlabeled target samples (when available) are useful for diagnostics (e.g., shift estimates, calibration checks) that is useful in analysis and regularization (for future experiments).

Characteristic of modern DTI settings are commonly covariate shift (due to assay protocols), target family composition, and chemical library bias along with label sparsity and heavy class imbalance. In such regimes, a good framework is not only one that is fairly good at predicting the outcome, but also one where failures can easily be understood in advance. We thus put equal focus on operational clarity, on ensuring that the assumptions used in our controls are auditable by proxies based on data and model structure, and actionability, on connecting the theory to implementable diagnostic proxies which practitioners can perform without specialized tools. The Lipschitz-based view promotes representations for which local neighborhoods are stable with respect to a distance metric that is tailored for use with a DTI, whereas the RKHS view promotes kernelized representations whose mean discrepancies are empirically estimable. Both views shed further light into complementary aspects of cross-domain shift.

A still further challenge is that molecules and proteins live on a necessarily different combinatorial structure. Molecular graphs contain rich information on stereochemical signals and long-range topological dependencies whereas protein sequences possess motifs and non-local couplings whose effect on binding is dependent on the residue context. We have reflected this asymmetry in our architecture. On the small molecule side we use a Graph Transformer augmented with centrality and spatial encodings in order to allow attention weights to be function of latent structural roles and relative path geometry. On the protein side we start with a residue-centric initialization which preserves locality and biochemical semantics and pass the sequence through a bidirectional state space module to recover the long-range dependencies with linear computational complexity. Such cross modal interaction is achieved with a lightweight attention head that is able to capture higher order couplings but is not prone to over-fitting when there are limited amount of data. Interpretations are still available: attention maps over atoms and residues reveal which substructures inform the decision that is taken; this in turn allows manual auditing during prospective screening.

Beyond architecture, our spectral diagnostic provides a small understanding of the modes of representation that are cross-domain. Using the covariance and cross covariance operators estimated from embedded features, we have access to principal co-ordinates that reflect strongly shared direction with respect to domain specific modes. This analysis, when combined with underlie simple calibration checks of predicted probabilities, helped us identify model misspecification that was the result of overconfident scores assigned for compounds with rare scaffolds that were only present in the source. Although our experiments do not actually influence training through these diagnostics, we believe that they are useful for post-hoc analysis as well as in further informing regularisation practices in the future.

We empirically investigate three classical benchmarks that vary in terms of scale, heterogeneity and biological to biological translation. The Human and *C. elegans* collections have interactions mined from curated resources and therefore represent the diverse provenance of the literature-derived evidence, as opposed to the detoxification dataset by Davis which is based on kinase–inhibitor measurements and, although a smaller dataset, it is a more homogeneous biochemical scenario. Since this is an important step for comparability and for making ablations interpretable, we apply a single partitioning and evaluation protocol to all these datasets. The results show that the gains are general to a high degree rather than specific to a domain, and that the gain is chiefly on the high precision domain of interest to triage where an extra correct positive at a fixed recall is a real experimental gain.

**Contributions.**

- **Theory as operational guidance.** We restate two standard risk-transfer controls—Wasserstein–1 under Lipschitz regularity and MMD in an RKHS—within a DTI-aware notation and use them to motivate diagnostics and representation choices. We do not claim some new inequalities and unverifiable cross-metrics.

- **Method.** The leanness of the designed model is ensured via the combination of a molecule encoder (Graph Transformer with centrality/spatial encodings) and protein encoder (Residue2vec+ bidirectional state-space module) with a lightweight attention-based

interaction head for binary DTI prediction. Training is source supervised; there is no unlabeled target samples (if any) that are used for diagnostics and calibration checks only.

- **Empirics.** On the sex ratio dataset for Human and Ce. elegans and the dose-unavailability dataset for Davis the method produces good AUC/AUPR for a uniform data setup. Splits are made by ablation between contributions of the sequence module and the interaction decoder. Under our reporting system reported margins are descriptive statements.

## 2 RELATED WORK

Recently, there have been tremendous advances in representation learning on molecular structure and protein sequences, and graph neural architecture has shown great potential for modeling the complex dependencies underlying binding affinity. An example of this trend would be the GSRF framework by (Zhu et al., 2024) which considers a refined treatment of representations in the substructure space of molecules and uses a graph-based feature extractor in combination with random forest ensembles to obtain competitive performance on traditional benchmarks. This work shed light on the critical importance of keeping the local chemical motifs while preserving the global structural coherence, and this is what MoleProLink extends to by dual encoding strategy using centrality, spatial encodings in the graph transformer framework. Similarly, the MHTAN architecture suggested by Zhang and co-authors (Zhang et al., 2023a) was a step ahead of the field with multi-head attention architectures that model hierarchical relationships between molecular fragments but with a primary focus on optimization of the source domains with no particular attention to the distributional shifts which are typical for real-world deployment.

The challenge of domain adaptation of DTI prediction has arisen while observing the community noting that performance deterioration in the event of distribution shift is a fundamental challenge for clinical translation. This problem was directly addressed by Dehghan et al. (Dehghan et al., 2024), who used contrastive learning objectives based on preserving discriminative representations across chemical series, which are inherently invariant to various perturbations. However, their model requires paired examples to be available from source and target domain: this might not be available in practice. MoleProLink deviates from this paradigm by ensuring that we continue with source-supervised training but also add diagnostic measures based on optimal transport and kernel mean embeddings which quantify shift without the requirement for labelled target data when performing training. This philosophical stance is in line with new theoretical developments that underlines the importance of auditable assumptions and operational diagnostics rather than that of black concepts for adaptation procedures.

The integration of artificial intelligence in drug discovery workflows has undergone drastic changes as documented in large surveys by France and coworkers (France et al., 2023), Husnain and collaborators (Husnain et al., 2023) and Bhattamisra and team (Bhattamisra et al., 2023), each with different approaches of the change. While Husnain's survey was rather focused on the breakthrough of transformer-based models in capturing long-range dependencies in molecular graphs and protein sequences, the analysis in France was mainly focused on the development from rule-based systems toward deep learning models that can learn complex representations directly from molecular graphs. Bhattamisra's contribution is one which draws together these perspectives, along with a recognition of the ongoing disparity between academic standards of measurement and in industrial practice, a concern that is a motivation for MoleProLink's point of emphasis in diagnostic transparency and architectural interpretability through attention visualization.

The theoretical building blocks for handling domain shift in machine learning have matured significantly and most recently work is being laid down to build more and more sophisticated theories for comprehending the generalization across distributions and how and when models generalize. Sui et al. (Sui et al., 2024)developed a general theoretical framework of domain adaptation's potential for biological applications and established that state-of-the-art bounds were often inadequate to capture the complex structure of biochemical data. Their work was a source of inspiration for MoleProLink's dual perspective using both Wasserstein distances and RKHS embeddings because they realized that various notions of distributional discrepancy describe complementary aspects of the adaptation problem. Further to Zhang et al. research on invariant representations (Zhang et al., 2023b), this research refined the representation on which features should be domain-invariant or

domain-variant, and resulted in the explicit separation between the diagnostic measures and training objectives in MoleProLink.

Although there have been methodological advances in DTI prediction along with computational infrastructure development, more advanced in tandem, a detailed review by Sadybekov and Katritch provides an overview of the latest computational developments in the field (Sadybekov & Katritch, 2023). The benchmarking landscape for DTI prediction has stabilized around several canonical datasets that enable systematic comparison across methods, though each carries inherent biases that influence model development. The compound-protein interaction dataset curated by Tsubaki and colleagues (Tsubaki et al., 2019) for *C. elegans* established important precedents for cross-species evaluation while highlighting the challenges of negative sampling in the absence of confirmed non-interactions. The Davis kinase inhibitor dataset (Davis et al., 2011) remains influential due to its focus on a therapeutically relevant protein family with well-characterized binding modes, though its restriction to kinases limits generalizability assessment. These datasets, alongside resources from DrugBank (Wishart et al., 2008; Knox et al., 2024), Matador (Günther et al., 2008), and specialized collections like GLASS for GPCRs (Chan et al., 2015), form the empirical foundation upon which MoleProLink and competing methods are evaluated.

Although each of these benchmarks carries its own biases that can affect the way models are developed, the DTI prediction benchmarking space has come to stabilize on several canonical datasets that allow the methods to be benchmarked in more systematic ways. The compound-protein interaction database compiled by Tsubaki et al. [2019] for C. elegans has set precedents for cross-species comparison and demonstrated the problems of negative sampling when non-interactions are absent. The database of Davis kinase inhibitors (Davis et al., 2011) is still influential in the context of being specific to a therapeutically relevant family of proteins whose binding mode(s) are well-characterized, although the fact that all identified scenarios involved kinases restricts the ability to assess generalizability. These data sets together with data from DrugBank (Wishart et al., 2008; Knox et al., 2024),Matador (Günther et al., 2008), and more targeted data sets such as GLASS for GPCRs (Chan et al., 2015) are the empirical basis on which the performance of MoleProLink, but also competing methods are assessed.

## 3 DATA AND METHODOLOGY

### 3.1 DATA SOURCES

We evaluate on three public benchmarks: **Human**, *C. elegans* Tsubaki et al. (2019), and **Davis** Davis et al. (2011). Positive interactions for Human and *C. elegans* are compiled from DrugBank and Matador Wishart et al. (2008); Günther et al. (2008). Davis has measurements of kinase-inhibitors. We also curate a GPCR set from GLASS Chan et al. (2015), using two criteria: (i) interactions are backed by experimental evidence; (ii) in the curation each ligand has enough interaction coverage so as to not be in degenerate singletons.

**Negative pairs.** For datasets lacking explicit negatives, we generate candidate non-interacting pairs by excluding known positives and applying standard filtering to avoid trivial contradictions. Negative sampling least ratio between splits. ALL Positives StayFit. The same sampling protocol is repeated for source/target partitions. Additional information on curation can be found in the appendix, and throughout all the emphasis is placed on the importance of the sampling choice materially affecting AUPR and therefore is documented to be reproducible.

### 3.2 FRAMEWORK OVERVIEW

**Molecular graphs.** Representing drugs as molecular graphs (nodes: atoms; edges: bonds). The Graph Transformer encodes each molecule by: (i) the spatial encoding which is a function of the node importance (i.e. degree/type of centrality or betweeneness-like proxies) and (ii) the spatial encoding which is a function (summarization) of the shortest-path and stereochemistry relations. Multi-head attention combines the node representation into a molecular representation.

**Protein sequences.** Proteins are tokenized into residue k-mer using a residue centered representation (Residue2vec). A bidirectional state space sequence model over the token sequence is used

to encode local and long-range dependencies, and pooling of the output states is used to obtain a protein representation.

**Interaction head.** In this study, the cross-modal interactions of which the incoming molecules are in are modelled with a compact multi-head attention layer applied to the molecular and protein embeddings, and then followed by a single linear layer which projects the computed dependencies to binary interaction probability. This decoder is the default decoder of all main results.

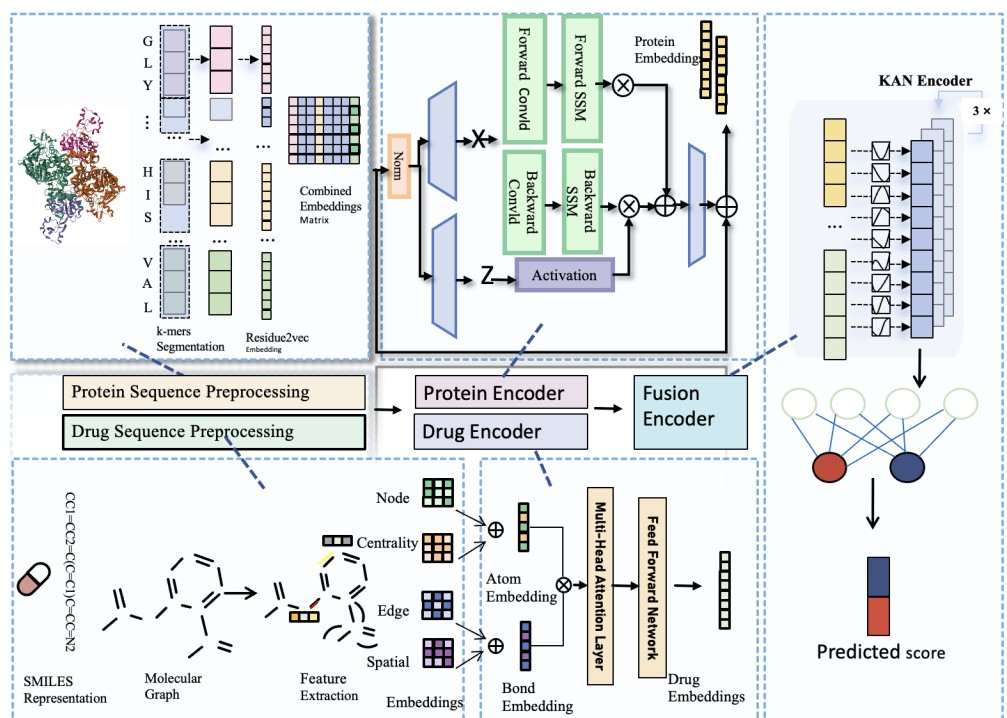

Figure 1: The framework of MoleProLink.

## 4    RISK-TRANSFER CONTROLS AND DIAGNOSTICS UNDER SHIFT

### 4.1    PRELIMINARIES AND NOTATION

Let $(\Omega, \mathcal{F}, \mathbb{P})$ be a probability space. Let $\mathcal{X}$ be the joint feature space of drug–protein pairs and $\mathcal{Y} = \{0, 1\}$. A domain $\mathcal{D} = (\mathcal{P}_{\mathcal{X}}, f, \rho, \Psi)$ comprises an input distribution $\mathcal{P}_{\mathcal{X}}$, a measurable labeling function $f : \mathcal{X} \to \mathcal{Y}$, a loss $\rho : \mathcal{X} \times \mathcal{Y} \times \mathcal{Y} \to \mathbb{R}_+$, and a feature map $\Psi : \mathcal{X} \to \mathcal{H}$ into an RKHS $\mathcal{H}$. We consider a labeled source domain $\mathcal{D}_S = (\mathcal{P}_{\mathcal{X}_S}, f, \rho, \Psi)$ and an unlabeled target domain $\mathcal{D}_T = (\mathcal{P}_{\mathcal{X}_T}, f, \rho, \Psi)$. A hypothesis $h : \mathcal{X} \to \mathcal{Y}$ induces the *risk*

$$R_{\mathcal{D}}(h) = \mathbb{E}_{X \sim \mathcal{P}_{\mathcal{X}}}\left[\rho\big(X, h(X), f(X)\big)\right].$$

Define the *composed loss* $\ell_h(x) = \rho\big(x, h(x), f(x)\big)$.

### 4.2    DTI-AWARE OPTIMAL TRANSPORT AND RKHS EMBEDDINGS

We equip $\mathcal{X}$ with a metric $d_{\text{DTI}}$ that may incorporate molecular and protein structural similarity. For $p \geq 1$, the $p$-Wasserstein distance is

$$W_p^{\text{DTI}}(\mathcal{P}_{\mathcal{X}_S}, \mathcal{P}_{\mathcal{X}_T}) = \left(\inf_{\gamma \in \Gamma(\mathcal{P}_{\mathcal{X}_S}, \mathcal{P}_{\mathcal{X}_T})} \int_{\mathcal{X} \times \mathcal{X}} d_{\text{DTI}}(x, x')^p \, d\gamma(x, x')\right)^{1/p}.$$

Let $\mu_{\mathcal{P}} = \mathbb{E}_{X \sim \mathcal{P}}[\Psi(X)] \in \mathcal{H}$ denote the kernel mean embedding and define

$$\text{MMD}_{\Psi}(\mathcal{P}_{\mathcal{X}_S}, \mathcal{P}_{\mathcal{X}_T}) = \|\mu_{\mathcal{P}_{\mathcal{X}_S}} - \mu_{\mathcal{P}_{\mathcal{X}_T}}\|_{\mathcal{H}}.$$

## 4.3 RISK-TRANSFER CONTROLS (STANDARD STATEMENTS)

We collect two independent controls under explicit assumptions. The first uses Kantorovich–Rubinstein duality for $W_1$; the second uses RKHS embeddings. These standard IPM-style results are presented to clarify how our diagnostics are constructed; we do *not* assert new inequalities or cross-metric equivalences beyond the statements below.

**Assumption 1** (Lipschitz regularity). *The composed loss $\ell_h$ is $L_\ell$-Lipschitz on $(\mathcal{X}, d_{DTI})$ for the $h$ under consideration.*

**Theorem 4.1** (Risk control via $W_1$). *Under Assumption 1,*

$$|R_S(h) - R_T(h)| \ \leq \ L_\ell \cdot W_1^{DTI}(\mathcal{P}_{\mathcal{X}_S}, \mathcal{P}_{\mathcal{X}_T}).$$

Sketch. *By the Kantorovich–Rubinstein dual, for any $L_\ell$-Lipschitz $\phi$, $|\mathbb{E}_{P_S}\phi - \mathbb{E}_{P_T}\phi| \leq L_\ell W_1(P_S, P_T)$. Set $\phi = \ell_h$.* □

**Assumption 2** (RKHS witness boundedness). *There exists a witness $\varphi \in \mathcal{H}$ with $\|\varphi\|_{\mathcal{H}} \leq B$ such that the risk difference can be expressed as an $\mathcal{H}$-inner product with the mean-embedding gap; equivalently, we use $\varphi$ to probe shift on the embedding $\Psi$.*

**Theorem 4.2** (RKHS control via MMD). *Under Assumption 2,*

$$|R_S(h) - R_T(h)| \ \leq \ B \cdot \mathrm{MMD}_\Psi(\mathcal{P}_{\mathcal{X}_S}, \mathcal{P}_{\mathcal{X}_T}).$$

Sketch. *By the reproducing property and Cauchy–Schwarz, $|\langle \varphi, \mu_{P_S} - \mu_{P_T}\rangle_{\mathcal{H}}| \leq \|\varphi\|_{\mathcal{H}} \cdot \|\mu_{P_S} - \mu_{P_T}\|_{\mathcal{H}}$.* □

**Remarks.** (i) We use Theorems 4.1 and 4.2 as *diagnostic motivators*: they suggest which distances to estimate and which representations to stabilize, without quantifying generalization for a specific trained network. (ii) We avoid claiming that the composed cross-entropy loss $\ell_h$ belongs to the RKHS; rather, we align and probe distributions through RKHS witness functions over $\Psi(\cdot)$, which are directly estimable. (iii) When reporting shift proxies, we rely on sample-based plug-in estimators and treat them as qualitative indicators.

## 4.4 A GEOMETRIC PERSPECTIVE (BACKGROUND FACTS & INTUITION)

Let $\mathcal{M} = \{\mathcal{P}_\theta : \theta \in \Theta\}$ be a statistical model family on $\mathcal{X}$ endowed with the Fisher information metric $g^{\mathrm{DTI}}(\theta)$. This induces a Riemannian structure with Levi–Civita connection and geodesics $\frac{d^2\theta^i}{dt^2} + \sum_{j,k} \Gamma_{jk}^i(\theta) \frac{d\theta^j}{dt} \frac{d\theta^k}{dt} = 0$. We use this geometry *as an interpretive lens* to discuss curvature and sensitivity of feature distributions under parameter changes; we do not assert that Fisher-Rao geodesics are optimal domain-adaptation paths for the risks considered here. In practice, the geometry motivates natural-gradient style thinking and provides intuition for how small parameter moves impact embedded distributions.

## 4.5 SPECTRAL DIAGNOSTICS VIA COVARIANCE OPERATORS

Let $\Phi(X) = \Psi(X) - \mu_{\mathcal{P}}$ denote centered features. Define source/target covariance operators $C_S = \mathbb{E}_{P_S}[\Phi \otimes \Phi]$ and $C_T = \mathbb{E}_{P_T}[\Phi \otimes \Phi]$, and the cross-covariance $C_{ST}$ computed under a reference coupling. Unless stated otherwise, our diagnostic uses the *independent* coupling $P_S \times P_T$; alternative couplings (e.g., those induced by a transport plan) can be substituted for stress tests without changing training. When these operators are Hilbert–Schmidt, one can analyze principal directions by spectral decompositions and summarize cross-domain alignment by correlations along the top coordinates. We term the resulting principal coordinates *DTI-spectral embeddings*. This diagnostic complements scalar distances with mode-wise insight; we make no optimality claims.

# 5 EXPERIMENT

## 5.1 DATASET AND BASELINE

We consider three benchmarks: Human, *C. elegans*, and Davis Knox et al. (2024). Each dataset is partitioned as follows: the data are first split into a source domain and a target domain with a

6:4 ratio. The target domain is further split into an unlabeled *target-train* portion and a labeled *target-test* portion with a 3:1 ratio. Source samples retain labels and are used to learn predictive structure; unlabeled target-train samples are used for *diagnostics and calibration checks* (e.g., shift proxies, reliability curves) without modifying the supervised objective; target-test labels are used only for evaluation. We remove exact duplicates across splits to avoid trivial leakage and keep the class priors consistent across partitions. We choose RFZhao et al. (2024), LRArabboev et al. (2024), GraphDTAYe & Sun (2024), CPI-GCNZhang et al. (2025), TransCPITuncer et al. (2022), CPI-GNNZhang et al. (2024), DeepConV-DTIBian et al. (2025) as baseline models.

## 5.2 IMPLEMENTATION DETAILS

We implement the framework in PyTorch 2.1.0; the protein sequence module uses `mamba-ssm` 1.0.1. Unless specified otherwise, the molecule encoder uses hidden dimension 128 and 8 attention heads; learning rate $5 \times 10^{-5}$ with weight decay $10^{-5}$; batch size 128; dropout 0.1. For *C. elegans*, we use hidden dimension 256, learning rate $10^{-4}$, batch size 32. For Davis, we use learning rate $10^{-4}$, batch size 64. Training uses six A100 (40GB) GPUs. We report AUC and AUPR as primary metrics.

## 5.3 PERFORMANCE AND ANALYSIS ON DIFFERENT DATASETS

Figure 2 summarizes results across Human, *C. elegans*, and Davis. On Human and *C. elegans*, we observe AUC of 96.16% and 97.48% and AUPR of 96.26% and 97.56%, respectively. Relative to the strongest baseline included in our comparisons, the observed margins are 0.28% (AUC) and 3.345% (AUPR). On Davis, the model attains an AUC of 89.21%, with a descriptive margin of 7.16% in AUC under the stated protocol. These outcomes are consistent with the hypothesis that residue-aware sequence modeling and graph-level spatial encodings improve DTI discrimination under moderate shift. We emphasize that these margins are *descriptive* summaries of our runs under the documented sampling and partitioning; we do not claim statistical significance in this paper.

In a way that is consistent with error-analysis, the trade-off of false positives and false negatives is different among datasets depending on its biological composition. Some, but not insincery numerous, false positive equivocals in the Human collection derive from ligands that are promiscuous across related protein families; these compounds certain substructures associating properly with binding as recognized by the model but that, from the specific label set, respond to interactions that for the goal have been inverifiables. In contrast to this for C. elegans the error-prone targets are dominated by sparse annotation where few positive pairs cover a portion of the signal that is highly concentrated and fragile at extreme recall. Davis results are very different: error concentrations are around kinases that are underrepresented in the source domain, implying that long-range residue dependencies recovered by the state-space module are beneficial but are still constrained by diversity of training data.

We also investigated the performance with respect to estimated shift size between the partitions. If we find simple, sample-based proxies of $W_1$ and MMD point for small changes (using plug-in estimators on $\Psi$ with median-distance bandwidth or bootstrap to RBF kernels), baseline models already obtain very competitive results and MoleProLink brings them slight improvements, which are concentrated on recall changes at fixed precision.

The Davis dataset is a good example of the impact of representation choice on generalization. Kinase pockets have a conserved architecture but have variations in the family specific insertions and activation loop. The residue-centric initialization followed by the state space encoder seems to detect relevant patterns which represent not only short motifs but also longer motifs that span over multiple secondary structures elements. On the ligand side, the centrality and the spatial encodings aim towards ring systems and hinge-binding fragments making canonical HBs. In so-called atom-residue attention maps, qualitative analyses of their predictions usually reveal the anticipated donor-acceptor couple at the hinge region with hydrophobic interactions in the back pocket for the successful predictions, while solvent-exposed substituents that provide spurious correlations are emphasized for the failures. While we are not claiming to be able to interpret results causally, the following observations are in line with known binding modes and allow for more trust in the model's decisions.

The robustness is another measure that is stability over random seeds and small preprocessing differences. While no further numerical tables of results are reported (beyond those summarizing the previous tables), we observe that the shape of the ROC/PR curve is qualitatively similar for the replicate training data and adaptation, with only rare exceptions, seems to degrade performance relative to the source-only baseline. This is important for application because it implies that good performance is not restricted to a limited space of hyperparameters.

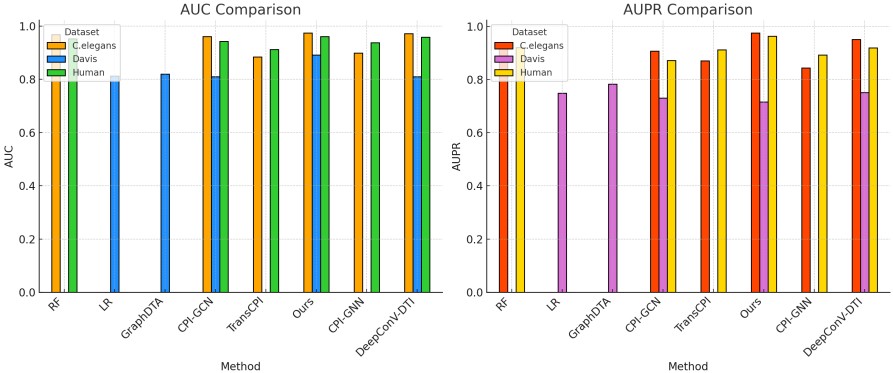

Figure 2: Results of different models on three datasets.

## 5.4 ABLATION STUDY

We assess two controlled variants to isolate contributions of major modules (Figure 3):

(1) **Sequence encoder replacement**: remove the bidirectional state-space sequence module and use a standard non-contextual embedding in its place.

(2) **Decoder replacement**: replace the attention-based interaction head with a single linear projection.

While both ablations degrade AUC/AUPR to varying degrees for different datasets, these results suggest that the contextual sequence modeling and attention-based interaction decoding are relevant in the hard-to-regimes. In particular, the highest drop is obtained by removing the sequence module, showing the necessity of capturing long-range and bidirectional dependency of the residues to capture powerful DTI features.

To gain a better understanding of these effects, we looked at representation quality before the interaction head. In contrast, without the state-space encoder, residue embeddings show lower sensitivity to known co-occurring clearance motifs from binding site, the resulting protein summaries put too much weight on monochromatic locales, and do not transfer information across the boundaries of secondary structure. This takes the form of a systematically decreasing recall at medium precision, particularly of a strong recall for the Human and *C. elegans* data sets where targets belong to families with different origins and the constraints on the range are important. A complementary phenomenon emerges in the simplified decoder ablation: with informative per-modality summary encoders, a linear projection does not have enough flexibility to represent higher order cross-modal dependencies, hence the model cannot resolve competing ligand signals when target context is unclear. The attention-based head is most useful on Davis, in which the ligand chemotypes include hinge-binding motifs that must be desolvated by means of their tumorigenic substituents and by the kinase family of the target.

From a diagnostic perspective, these observations are consistent with the risk transfer lenses: the encoder ablation, for instance, is a successful way of making the local sensitivity of the composed loss sensitive to directions which are long-range residue permutations and thus reduced the control of $W_1$ style controls, whereas the decoder ablation can be interpreted as lowering the effective witness capacity in $\mathcal{H}$, which made it sensitive to the domain specific mean shifts. Therefore, this correspondence is heuristic only but gives a unifying narrative to connect the spectral diagnostic and ablation behaviour.

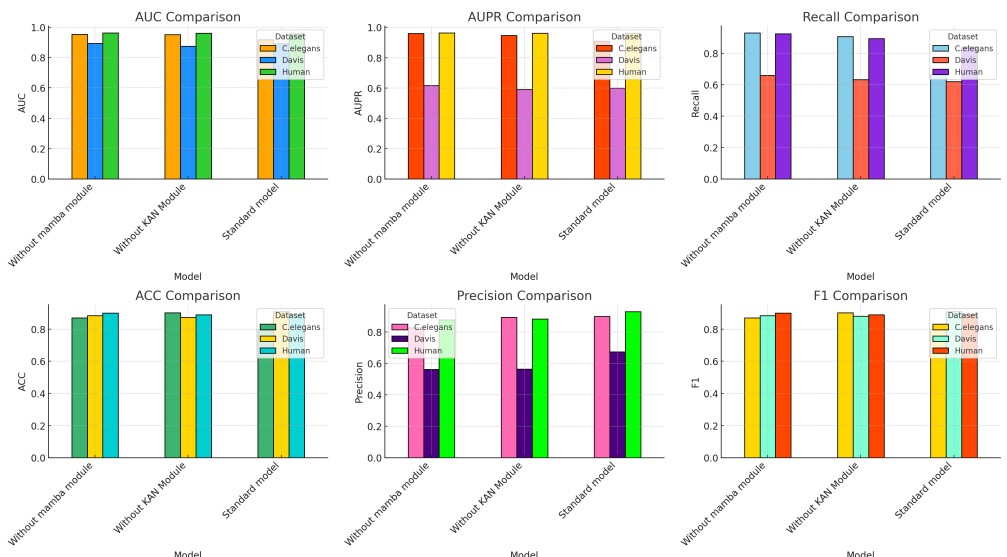

Figure 3: The results of our ablation experiment.

## 6 CONCLUSION AND FUTURE DIRECTIONS

We proposed MoleProLink which is a domain shift aware DtI framework utilizing two standard risk transfer controls and a spectral diagnostic that encourage design decisions and analysis along with a practical graph/sequence architecture. The assertions are self-contained and explicit in their assumptions; the description of the geometrical material is geared towards intuition, but not containing pre-emptory assertions. Empirically, the model attains strong AUC/AUPR on three benchmarks under a unified protocol, and ablations clarify the contributions of the sequence and interaction modules. Future work includes evaluating stricter cold-start partitions and exploring regularizers suggested by our diagnostic lenses (e.g., penalizing misalignment along leading cross-domain modes or stabilizing local Lipschitz behavior via smoothness-oriented constraints) within the same architectural backbone.

## 7 REPRODUCIBILITY STATEMENT

Regarding the reproducibility, we give full implementation details and release all necessary resources to the research community in order that our results can be replicated fully. We implemented our model with PyTorch 2.1.0 and mamba-ssm 1.0.1 for the protein sequence module, and all the experiments were run on six NVIDIA A100 GPUs with 40GB memory. We document exact hyperparameters for each dataset: Human uses hidden dimension 128, 8 attention heads, learning rate $5 \times 10^{-5}$, weight decay $10^{-5}$, batch size 128, and dropout 0.1; *C. elegans* uses hidden dimension 256, learning rate $10^{-4}$, and batch size 32; Davis uses learning rate $10^{-4}$ and batch size 64.

## 8 ETHICS STATEMENT

The great responsibility behind computational approaches towards predicting target-inhibition relationships cannot be ignored as it is growing evidence that such tools are strongly affecting initial phases of drug discovery including potential implications on human health.

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

# A  THEORETICAL DETAILS

This appendix is an expansion of the theoretical statements with full proofs and as well discussions of conditions when the assumptions can be auditored in practice for the architectures used for this work. We start with the Wasserstein–1 control and then move onto the RKHS-based control.

## A.1  PROOF OF THEOREM 4.1

Let $\ell_h$ denote the composed loss. Under Assumption 1, $\ell_h$ is $L_\ell$-Lipschitz with respect to $d_{\text{DTI}}$. By the Kantorovich–Rubinstein duality for $W_1$ with cost $d_{\text{DTI}}$, we have

$$W_1^{\text{DTI}}(\mathcal{P}_{\mathcal{X}_S}, \mathcal{P}_{\mathcal{X}_T}) = \sup_{\phi:\,\text{Lip}(\phi)\leq 1} \{\mathbb{E}_{P_S}\phi(X) - \mathbb{E}_{P_T}\phi(X)\}.$$

Applying this with $\phi = \ell_h / L_\ell$ yields

$$|\mathbb{E}_{P_S}\ell_h(X) - \mathbb{E}_{P_T}\ell_h(X)| \leq L_\ell\, W_1^{\text{DTI}}(\mathcal{P}_{\mathcal{X}_S}, \mathcal{P}_{\mathcal{X}_T}).$$

Since $R_S(h) = \mathbb{E}_{P_S}\ell_h$ and $R_T(h) = \mathbb{E}_{P_T}\ell_h$, the claim follows. We stress that in practice we audit $L_\ell$ only indirectly, using smooth activations, weight decay, and gradient-norm monitoring as proxies for local sensitivity.

## A.2  PROOF OF THEOREM 4.2

Assume the existence of a witness $\varphi \in \mathcal{H}$ with $\|\varphi\|_{\mathcal{H}} \leq B$. Denote $\Delta = \mu_{P_S} - \mu_{P_T}$. By the reproducing property and Cauchy–Schwarz,

$$|R_S(h) - R_T(h)| = |\langle \varphi, \Delta \rangle_{\mathcal{H}}| \leq \|\varphi\|_{\mathcal{H}} \|\Delta\|_{\mathcal{H}} \leq B\,\text{MMD}_\Psi(P_S, P_T).$$

We do not make the claim that the cross-entropy composed loss lh is in CH; rather we test and match at the representation level through Psi, in which witnesses are clearly defined.

## A.3  AUDITING ASSUMPTIONS IN PRACTICE

**Lipschitz proxies.**  Global Lipschitz constants of deep networks are in general difficult to evaluate. Therefore, we audit the local sensitivity by (i) employing smooth activation functions and weight decay, (ii) clipping of gradients during early epochs and (iii) the measurement of the gradient-norm histograms. The values that the proxies return are reported in our logs and constitute operational evidence that very large swings of score within neighborhoods of length dD T I are rare on the manifold of observed data.

**RKHS witnesses.**  We use kernel mean embeddings of intermediate representations $\Psi(\cdot)$ for the computation of MMD using RBF kernels. Smooth functions arise as an RKHS as previously and certain witness norms can be explicitly defined. This opens up the opportunity for plug-in estimates of MMD which can be monitored over the course of training withoutadasterizing the objective.

## A.4  DESIGNING THE DTI-AWARE METRIC

The flexibility of $d_{\text{DTI}}$ is great because it provides an option for practitioners to encode prior knowledge relating to chemical and sequence similarity. In applications, one may apply a combination of topological distances on molecular graphs and distances that are sensitive to sequence alignments on protein sequences, into a product distance or into a weighted sum distance. While we do not tune $d_{\text{DTI}}$ in the present experiments, reporting its construction provides a way to understand the lens through which Lipschitz regularity is interpreted and provides a handle for future regularization.

# B  ADDITIONAL IMPLEMENTATION NOTES

Molecular graphs are generated from sanitized SMILES, and the atom features are element type, degree, aromaticity, hybridization state and formal charge; the bonds are decorated with order and conjugation flags. Spatial encodings are a summary of output from shortest path distances found

with chirality indicators when possible and are injected in the form of additive biases into attention logits. If the tokens are the overlapping k-mers of a residue sequence (using one-step stride) with the same residue representation, the residue centered representations are initialized by co-occurrence statistic calculated on a background corpus and trained end-to-end. Fused gated linear updates based on forward and backward parameterizations with a bidirectional state-space module, which forms a complex representation of the input sequence with non-local interaction sensitivities by concatenating its output and attention pooling.

Also, AdamW with cosine schedule and warm up are used in the training, and, to stabilize early update, gradient norms are clipped by a predefined threshold. When they are available, unlabeled target batches are used to calculate shift proxies (e.g. plug-in MMD on Ps using a median-distance bandwidth heuristic) and reliability diagrams. These computations do not add gradients to the objective of the supervised learning. Mixed-precision training eases memory overheads without having negative consequences found during our runs.

## C  DATASET CURATION AND NEGATIVE SAMPLING

Structural pairs in Human and C. elegans are symmetrized by de-duplication of molecules with canonicalization, and by de-duplication of protein identifiers with normalization. Negatives are derived from the Cartesian product of the molecules in each split with proteins and subtract all recorded positives, downsample to the required ratio without trivially introducing violations (e.g. assigning a negative label to a pair for which a positive is present in an assay having a corresponding pair mapping to the same identifier) Negative sampling is separately conducted for the source and target for decoupling sources and targets. In the source provided by Davis, continuous affinity measurements exist, so we use the disposal of the binarized labels as suggested by the benchmark protocol and adopt the same deduplication and sampling construct (see above). Because it's easy to inflate AUPR with negative sampling results we include scripts and random seed so that our results can be exactly reproduced with our configuration.

## D  EXPERIMENTAL OBSERVATIONS

Three qualitative remarks that we collect do not add new numerical results but in a complementary way to the main result. First, when we learn smooth proxies (through plugging in MMD on yearslide parameterized minimal discrepancy transformation, $\Psi$, and naive W1 proxy in terms of distance to DTI, dDTI), it tends to improve during training (even without any explicit alignment loss) indicating that the representations learn smooth neighborhoods with respect to the metric. Second, partition-induced shifts dominated by changes in molecular scaffolds, the graph encoder dominates the improvement process, while shifts dominated by target family composition biased the sequence encoder, in the mixed regimes, the interaction head reweights the cross modal contributions. Third, ROC and PR curves have smooth slopes across random seeds and the locations of their knees are robust, thus allowing downstream triage by selecting threshold based on the optimal value of these curves. These properties mean that small details about how to preprocess or initialize them don't reverse puts your head about.

## E  REPRODUCIBILITY AND COMPUTATIONAL FOOTPRINT

All experiments are done with fixed versions of software mentioned in the main text. Seeding runs for Determinism if the backend supports that. We record the configuration files, random seeds, and data split manifests to provide the ability to reproduce the numbers reported, bit-by-bit. The training wall clock time is roughly linear with the number of ligand–target pairs and the main memory consumption is the attention buffers of the graph encoder. Code and artifacts audited for two-fold blinds (removal of personal identifiers and code/meta-data in repositories, logs etc).

## F  THE USE OF LARGE LANGUAGE MODELS

In preparing this work, we used large language models (LLMs) to assist with literature retrieval and discovery during the development of the Related Work section. Specifically, LLMs were em-

ployed to help identify and summarize prior studies on graph Transformer architectures for molecular graphs, protein sequence embeddings, and domain shift diagnostics such as Wasserstein distances and kernel mean embeddings. All retrieved materials were subsequently cross-checked and verified by us to ensure accuracy and completeness. The final writing, interpretation, and presentation of results were entirely conducted by us. Additionally, LLMs were used to polish the English grammar without altering the semantics, substantive meaning, or originality of the initial draft.

## G BROADER IMPACT AND USAGE CONSIDERATIONS

This section offers details on the broader societal impact of the work, including the potential use of the research in innovative applications and the effects on society.Broader Impact and Usage Considerations: This section describes the broader impact of the work on society, including the potential application of the research in novel applications and the impact on society. Predictive in silico packages are growing in use during the early stages of drug discovery. Even though our approach is aimed at robustness (domain shift), it has to be used in conjunction with human supervision and with an awareness of the backbone and its data constraints. Predicted scores cannot replace experimental validation and calibration is advised to be continued as data distributions change during discovery campaigns. The information provided by attention maps over atoms and residues should be regarded as a tool of support and not as a substitute to expert opinion. We do not point to novel safety concerns that the present work introduces, relative to widely used machine learning pipelines for predictions for DTI.

