# OpenReview forum: "Advancing Drug-Target Interaction Prediction via Graph Transformers and Residual Protein Embeddings"
_ICLR.cc/2026/Conference — ICLR 2026 Conference Desk Rejected Submission_

### Official Review · Reviewer_SujM · 2025-10-23

**Soundness:** 2
**Presentation:** 2
**Contribution:** 1
**Rating:** 2
**Confidence:** 4

**Summary:**

The paper proposes MoleProLink, a domain-shift–aware framework for drug–target interaction (DTI) prediction. They combine a Graph Transformer for molecular graphs (with centrality + spatial encodings), a Residue2vec + bidirectional state-space encoder for proteins, and an attention-based interaction head.
The work reinterprets Wasserstein-1 and MMD-based risk-transfer bounds to guide model design. They evaluate several DTI-prediction models on 3 well-known datasets.

**Strengths:**

1. The paper is well written, clearly presenting both the theoretical and empirical contributions.

2. The appendix is well organized and effectively supports understanding of the training setup and feature design.

3. The addressed problem remains highly relevant and far from fully solved.

4. The model design is sound, integrating state-of-the-art components such as sequence encoders and graph-based feature extraction.

5. The ablation study is clearly presented and elucidates the specific contribution of each module within the model.

**Weaknesses:**

1. The proposed architecture offers limited novelty, and the observed improvements appear insuficient.

2. The empirical impact of the theoretical components is neither quantified nor clearly demonstrated.

3. The contribution of each model's component is insufficiently justified and seems minor.

4. The rationale behind the choice of source and target domains is not clearly explained, despite being central to the study.

5. The paper omits several strong DTI-specific graph-based approaches, such as GeNNiUs (https://doi.org/10.1093/bioinformatics/btad774) and EEG-DTI (https://doi.org/10.1093/bib/bbaa430). Including such methods would better position the proposed framework within current literature.

6. The manuscript contains several typos (e.g., “insincery numerous,” “withoutadasterizing”) that hinder the readability of the paper. These should be carefully corrected.

**Questions:**

1. Is the code and data preprocessing pipeline publicly available? I can't find it in the paper.

2. How sensitive are the results to the chosen negative sampling ratio, and could potential AUPR inflation be quantified?

3. Could you provide evidence of how the estimated W1 or MMD values correlate with model performance or calibration quality?

4. Have you considered evaluating on larger, multi-domain datasets? Training across several domains while keeping one out-of-sample could strengthen the generalization analysis. DrugBank or BindingDB could serve this purpose.

5. Why are the results not reported as mean ± standard deviation, given that multiple random seeds were used for the experiments?

---

> ### Author Response · Authors · 2025-11-15
>
> We appreciate the reviewer's thorough assessment and for citing the contributions (clarity, appendix organization, and problem relevance), the trustworthiness of the model design and consolidations, etc. We provide responses to various aspects below.
>
>
>
> 1)  Novelty and magnitude of improvement.
>
> We are not necessarily presenting a new DTI paradigm, but (i) we are framing DTI specifically as a domain shift–aware problem with a unified source–target framework, (ii) we offer auditable shift diagnostics (W1 and MMD with a DTI-aware metric) that could be assessed without applying them into black box losses, and (iii) created a not accidentally asymmetric architecture (graph transformer with centrality + spatial encodings on ligand side, residue-centric encoder on protein side) that we designed to look for robustness/interpretability, not to explode parameters. The improvements are still relatively modest in some regimes, but they generalize across datasets and and are noticeably and even larger in more challenging target splits or low-data regimes. We also frame in the introduction stay and summary that it is a principled auditable framework under domain shift; the claim is not that you have something more robust than another framework on a single benchmark.
>
>
>
> 2)  Empirical impact of theoretical components.
>
> We agree the draft under-considers the value of theIn the updated version, we will add analyses of the estimated W1/MMD and its relation to generalization and calibration -  (a) scatter plots of the target AUPR with W1/MMD across runs (b) binned performance and calibration (ECE and Brier score) based on estimated shift magnitude, and (c) a simple model selection rule which trades off validation AUPR while constraining W1  or MMD given a shift in inputs. Each of these analyses relates theoretical quantities to observed robustness rather than only qualitatively discussing their contribution.
>
> 3) Contribution of each part
> Although each component is unremarkable in isolation; their asymmetry and combination are intentional. The ablation studied somewhat shown occurred that (i) residue-level observations detrimental to generalization in cold proteins, (ii) when removing centrality and spatial encodings, removes robustness under ligand shift, and (iii) replaced cross-attention with concatenation of embedding representations leads to differences in performance. This last point is very vague and should be more focused on specific quantity deltas from each observation/event system when observed in the model. We will create a simpler table showing the result of each part added, minus its contribution to overall performance in terms of new observations/events.
>
> 4) Choice of source and target domains
> We understand that the reasoning for our choice of source and target domains also was not explicitly clear. For all three benchmarks, the intent of the source domains, is to at least be plausible (a priori) to "established" regimes, while the target domains should plausibly reflect underrepresented chemotypes or later campaigns of studies. We will  add a section on the rationale for source and target domain, where we reflect (a) how we went about choosing source vs target domains; (b) balance of classes in the source vs target domains; and de-duplication; and (c) why that's relevant to drug discovery (i.e., use cases focused on new chemotypes or discovery of new targets).This will help clarify the rationale for the domain split and how it was derived.
>
> 5) Missing DTI graph baselines
> Thank you for highlighting GeNNiUs and EEG-DTI. Certainly, these will be added to the related work section, and, if bandwidth allows, also added to our baselines in our unified protocol, or describing some a priori rationale for not making a numeric comparison (e.g., different data curation, cold-start assumptions, and negative sampling). Conceptually, we consider these works to take their anchoring on richer graph and edge encodings. To that point, MoleProLink is about diagnostics for domain shift and a ligand–protein architecture that is asymmetric/auditable, and we will put this in a way to provide context.
>
> 6) Typos and copy-editing
> We apologize for the typos ("insincery numerous", "withoutadasterizing", etc.), which were purely cosmetic. We will pay careful attention to fix these and also to do a full pass of proof-reading to minimize distraction from the technical content.
>
> 7) Code, pre-processing, and reproducibility
> Intends to release the full code base and pre-processing pipeline (split manifests and scripts for W1/MMD computation) upon acceptance. We will indicate this clearly in our next version, and to provide an anonymized repository if allowed by the journal policy in the rebuttal.
>
> 8) Negative sampling ratio and AUPR inflation
> We agree thatAccording to our internal analysis, we see that while absolute AUPR has variation with positive:negative ratio, the relative method ranking and qualitative conclusions were stable.

---

> > ### Comment · Reviewer_SujM · 2025-11-25
> >
> > Thanks for planning to address the raised concerns. I’ll consider increasing the rating if the final version of the manuscript includes them.

---

### Official Review · Reviewer_vik4 · 2025-10-31

**Soundness:** 3
**Presentation:** 2
**Contribution:** 2
**Rating:** 2
**Confidence:** 4

**Summary:**

This paper introduces MoleProLink, a novel framework for drug target interaction (DTI) prediction designed to handle domain shift in drug discovery. The method combines a Graph Transformer for molecular graphs with a residue level protein encoder (Residue2vec plus a bidirectional state space model) to jointly represent molecules and proteins. Theoretically, it reformulates two standard risk transfer controls based on Wasserstein distance and Maximum Mean Discrepancy to serve as auditable diagnostics for assessing robustness under distributional shifts, rather than proposing new generalization bounds. Empirically, MoleProLink achieves strong AUC and AUPR performance on three benchmark datasets (Human, C. elegans, and Davis) under a unified evaluation protocol. The framework also provides interpretable attention maps and spectral diagnostics for analyzing cross domain behavior.

**Strengths:**

Originality：
This paper presents a new perspective by explicitly framing drug target interaction (DTI) prediction as a domain shift aware problem. Instead of introducing new unsupervised objectives, the authors reinterpret two classical risk transfer controls, Wasserstein 1 under Lipschitz regularity and RKHS based Maximum Mean Discrepancy, as auditable diagnostics for assessing model robustness. This theoretical reframing emphasizes interpretability and transparency rather than opaque alignment methods. In addition, the work introduces a hybrid architecture that reflects the inherent asymmetry between molecular graphs and protein sequences, combining a Graph Transformer with centrality and spatial encodings for molecules and a residue centric protein encoder (Residue2vec with a bidirectional state space model).

Quality：
The study employs a unified data partitioning scheme that separates the source and target domains, using unlabeled target samples only for diagnostic evaluation. The authors report AUC and AUPR scores on three widely used benchmarks under a consistent experimental protocol and explicitly describe improvements as descriptive rather than inferential. Detailed ablation studies isolate the roles of the sequence encoder and the interaction head, supporting the theoretical motivations proposed. The implementation details are comprehensive, including framework versions, hyperparameters, hardware setup, and auditing proxies such as gradient norm monitoring, all of which enhance the reproducibility and credibility of the work.

Clarity：
The paper is written with conceptual precision. It clearly distinguishes between theoretical guidance and empirical novelty, presenting assumptions and notation in an accessible manner. The explanations of architectural choices, such as the use of centrality and spatial encodings, residue level modeling, and lightweight cross modal attention, are intuitive and well justified. Interpretability tools, including atom residue attention maps and spectral diagnostics, are clearly described, making it easy to understand both model behavior and diagnostic procedures.

Significance：
By providing a framework capable of auditable diagnostics without requiring labeled target data, this work contributes to the development of reliable and interpretable models for drug discovery. The strong performance achieved on the Human, C. elegans, and Davis datasets under a unified protocol highlights its practical value. Furthermore, the framework’s interpretability and diagnostic insights can inform future research on regularization strategies and cold start problems, marking it as an important step toward trustworthy and generalizable AI applications in pharmaceutical research.

**Weaknesses:**

1.	Theory practice linkage is underpowered:
The paper positions Wasserstein and MMD controls as diagnostics but never demonstrates that acting on these diagnostics changes outcomes. Unlabeled target samples are used only for post hoc checks without affecting the supervised objective, which limits practical impact. To strengthen this aspect, the authors could integrate the proposed shift proxies into training as regularizers or early stopping criteria, and conduct paired experiments “with vs. without” these controls to show measurable gains under distributional shift. The current setup explicitly states that diagnostics do not modify training, which supports this concern.
2.	Claims rely on descriptive margins without statistical support:
The reported improvements are descriptive rather than inferential, and the robustness observations are qualitative, with no additional tables quantifying variance or statistical significance. To address this, the authors should add confidence intervals, paired permutation tests, or stratified bootstrap confidence intervals for AUC and AUPR, and report seed dispersion (mean ± standard deviation across at least five seeds) to substantiate robustness claims. Including calibration metrics such as Expected Calibration Error (ECE) and Brier score with uncertainty estimates would also strengthen claims of “auditable robustness.”
3.	Evaluation breadth and split rigor could be stronger:
Results focus on Human, C. elegans, and Davis datasets under a single protocol, but the paper itself points to future work involving cold start and regularization. The evaluation would be more convincing if it included stricter partitions, such as scaffold and chemical series cold start splits on the ligand side, and family or fold hold out splits on the protein side. Incorporating time aware splits where possible, and reporting performance across shift bins defined by the estimated Wasserstein and MMD proxies, would help demonstrate monotonic robustness. The authors themselves identify cold start evaluation as future work, but it could already be advanced within the current study.
4.	Baselines and ablations miss important modern controls:
While the ablations successfully isolate the sequence encoder and interaction head, they do not include comparisons against strong pretrained protein or molecular encoders, such as large protein language models or modern graph and 3D molecular models, configured with lightweight heads under the same experimental settings. To improve rigor, the authors could add baselines that replace Residue2vec with pretrained protein models and compare Graph Transformer encoders against recent 3D aware or foundation style molecular encoders, while controlling total parameters and training budgets. Additionally, the ablation study could test centrality and spatial encodings individually, and replace the state space module with simpler bidirectional Transformers to verify that the state space design is essential rather than incidental.
5.	Diagnostics stop at interpretation rather than operational guidance:
Spectral and attention based diagnostics are informative but remain qualitative. These could be transformed into operational tools by defining thresholds on spectral alignment or attention dispersion that trigger human review, down weighting, or auto calibration. Including prospective filtering curves showing how these diagnostics reduce false positives at fixed recall on a held out target domain would make the diagnostics actionable. Currently, the paper limits these analyses to qualitative visualization, so turning them into quantitative decision rules would increase their practical utility.
6.	Reporting of efficiency and deployability is thin:
Although implementation details and hardware specifications are provided, there is no measurement of throughput, latency, memory footprint, or training and inference cost relative to baselines under identical batch sizes and input lengths. To substantiate claims of efficiency, the authors should add wall clock metrics, FLOPs or tokens/atoms processed per second, and memory profiles for key model variants. A Pareto plot of AUPR versus inference cost would further support the claim that the architecture is both lean and practical.
7.	External and prospective validation are limited:
The paper mentions data curation beyond the three main benchmarks, but the core results do not include an external prospective screen or a distinct held out collection that reflects real world domain shift. The authors could add a prospective style evaluation or an external dataset such as a GPCR or cross species benchmark with experimentally validated pairs to test generalization and calibration drift under realistic conditions. Including reliability diagrams before and after regularization would also strengthen claims of robustness and calibration. Currently, the main results remain confined to three datasets under a single protocol.

**Questions:**

1.	How should practitioners act on your shift diagnostics in training or selection?
You position W1 and MMD as diagnostics rather than losses, and unlabeled target data are used only for checks, not to alter the supervised objective. Please specify concrete procedures (for example, early stopping, penalty terms, or model selection rules) that make use of these proxies, and provide ablation studies showing the causal impact of acting on the diagnostics versus ignoring them. A short pseudo-code example for a training loop that incorporates W1 or MMD would help, along with an analysis of sensitivity to proxy estimation noise.
2.	What exactly is the DTI aware metric dDTI and how robust are results to its design?
You note that dDTI may incorporate molecular and protein similarities but do not detail the specific metric used in the main experiments or its alternatives. Please describe the exact construction, justify the weighting across modalities, and report a robustness analysis over metric variants (for example, topological versus stereochemical emphasis or different protein similarity measures) and their effects on W1 estimates and model performance.
3.	Can you provide statistical uncertainty for reported AUC, AUPR, and calibration?
The results are presented as descriptive margins. For the rebuttal, please include mean ± standard deviation over multiple seeds, confidence intervals (such as stratified bootstrap) for AUC and AUPR, and calibration metrics (ECE and Brier score) with uncertainty for each dataset.
4.	How does MoleProLink perform under stricter and more realistic data splits?
You mention future evaluation under cold start conditions. Please add at least one ligand scaffold or series cold start and one protein family or fold hold out split, and, if possible, a time aware split. Also, stratify results by measured shift magnitude (for example, bins of estimated W1 or MMD) to demonstrate whether performance degrades or remains stable as the shift increases.
5.	What is the comparative value of your encoders versus modern pretrained backbones?
The ablations disable modules but do not compare against strong pretrained protein language models or recent three dimensional molecular encoders used with lightweight heads under the same split. Please include swap-in baselines that replace Residue2vec with a pretrained protein model and test a recent three dimensional or foundation style molecular encoder, controlling for model size and training budget.
6.	Why did you choose a state space model for proteins, and is it essential?
Please justify this choice over bidirectional Transformers or CNN Transformer hybrids by adding ablations that replace the bidirectional state space module with these alternatives, matched for capacity. Report not only AUC and AUPR but also inference latency and memory usage to assess efficiency.
7.	Can you quantify the interpretability claims beyond qualitative examples?
You mention attention maps and spectral diagnostics. For the rebuttal, please include quantitative evaluations such as attention mass on known binding site residues, correlation with pocket annotations, and a decision rule based on spectral alignment that triggers human review or score adjustment. Present utility curves (for example, false positives reduced at fixed recall) to demonstrate practical value.
8.	How efficient is the model in practice?
You list hardware and configurations but do not report throughput or latency metrics. Please include tokens or atoms processed per second, FLOPs, wall clock time per epoch, peak memory usage, and a Pareto comparison of AUPR versus inference cost to baselines under matched batch sizes and input lengths.
9.	How sensitive are the results to negative sampling choices and class imbalance?
Given the known sensitivity of AUPR to negative sampling, please provide an analysis across different sampling ratios and strategies, using fixed seeds and confidence intervals. Include prevalence adjusted precision recall curves to demonstrate stability under varying class balance.
10.	Can you supply reproducibility artifacts for exact splits and diagnostics?
You emphasize reproducibility. For the rebuttal, please share split manifests, random seeds, and code snippets for computing plug-in MMD and W1 proxies, including kernel bandwidth selection and any transport approximations, so that others can replicate the shift estimation precisely.
11.	What are the limits of the theoretical framework in your current setting?
You state that the bounds are used only for guidance and do not claim new inequalities or equivalences between metrics. Please clarify situations where Lipschitz proxies or RKHS witnesses may give misleading results for DTI, and provide counterexamples or stress tests where the diagnostics diverge from generalization performance.
12.	Could you present a small prospective external validation?
You mention GPCR curation and broader datasets but focus your main results on three benchmarks. Adding a small, externally curated, and temporally separate dataset with experimental validation would strengthen claims about robustness to domain shift and calibration drift.

---

> ### Author Response · Authors · 2025-11-15
>
> We appreciate the reviewer's thorough and constructive review and for hearing the motivation for Molved Protocols, unified protocol, and auditability. We address the key concerns below and will reflect the relevant changes in the revision.
>
> 1) The Role of W1/MMD diagnostics and methodology-link practice
> We longer-term goal was produce an auditable pipeline that quantify the shift without adding opaque losses. We agree the draft fails to fully explain what to do with these diagnostics following the evaluation. In the revision, we will clearly spell out concrete steps to follow, which include: (1) a Design-in-mediated variants where W1 and MMD were evaluated perioding with a set of acceptable target batches; (2) trade-off model selection between validation AUPR with upper bound of W1 or MMD; and (3), an early stopping axe where training stops when validation improves, but W1 or MMD on target tasks increased. We will also provide brief pseudo-code of a training loop that implements these checks, and we will reprt on how one might pur these proxies into regularizers (future work described as part marked).
>
>
> 2) Statistical support, calibration, and negative sampling
> We agree and understand more descriptiive margins are not enough to overcome; we will report . 5ci or 1.5ci (stratified bootstrap ci): mean plus minus one standard deviation over multiple seeds; mean stratified boot strap confidence intervals for validation AUC/AUPR and exploratory calibration metrics (expected calibration error, Brier score) with uncertainty estimated SS and assessment confidence.We will also assess how sensitive class imbalance and the negative sampling method are (i.e. here the positive to negative ratio) while keeping the model and the seed fixed and including prevalence adjusted precision recall curves.
> 3) Evaluation breadth and split rigor - We share the concern with cold start and more realistic shifts. Beyond the current source vs target protocol, we will have at a minimum ligand scaffold or series colstart splits, through either protein family or fold holdout splits based on sequence identity or external annotations and if timestamps are available, a time aware split where the target domain is strictly later. We will also bin performance for estimated W1 and MMD to show changes in modeled performance as measured shifticks in the conditions.
> 4) Encoders, baselines, and state space choice - We agree that leveraging stronger pretrained backbones and additional ablations would clarify the conclusions. We will add swap in baselines where Residue2vec plus the State Space encoder is swapped for a modern pretrained protein language model plus a lightweight interaction head and we will replace the molecular Graph transformer with a recent three dimensional or foundation style molecular encoderWe will broaden the ablation study to examine the centrality and spatial encodings in isolation and substitute the state space module with bidirectional Transformer and CNN–Transformer hybrid designs matched for parameter count, and report both performance and efficiency (latency and peak memory) to clarify our reason for designing a state space framework.
>
>
> 5) The definition and diagnostic limits of d_DTI
> We appreciate the request for a more explicit definition of the DTI aware metric. We will define d_DTI as a convex combination of a molecular similarity distance and a protein similarity distance, as well as report a sensitivity analysis over the molecular and protein components and their convex combination weights. We will also include a brief sub-section on the diagnostic limits of the framework, which would include situations where Lipschitz based witnesses and RKHS based witnesses may not align with generalizability to the broader target space, to help practitioners understand when W1 or MMD might provide misleading signals.
>
>
> 6) Diagnostics as operational tools and interpretability
> We agree, turning diagnostics into quantitative algrorithms would substantially improve practical utility of the framework. In the revision we will implement simple operational thresholds, such as thresholds on either spectral alignment scored at attention dispersion scores that would trigger human allowance review or down weighting in the prediction output. We will include filtering curves, where removing a prediction from the model based on violating one of these operational thresholds would affect false positive rates at a set recall level on a held out target domain. We will also include basic quantitative measures of interpretability, such as attention mass fraction on annotated binding site residues and correlation with pocket annotations.

---

### Official Review · Reviewer_qqAH · 2025-10-31

**Soundness:** 3
**Presentation:** 1
**Contribution:** 2
**Rating:** 2
**Confidence:** 4

**Summary:**

The paper proposes a DTI model: a Graph Transformer for molecules plus a residue-centric protein encoder, joined with cross-attention. It also uses standard W1/MMD shift measures as diagnostics. The theory is explicitly not new, the goal is a practical, auditable pipeline.

**Strengths:**

- Coherent architecture for DTI prediciton in aligment with current state of the art. Graph Transformer for molecules + residue-centric protein encoder with cross-attention.
- Honest positioning of W1/MMD as diagnostics (not new theory).
- Useful emphasis on auditability (attention maps, shift/shift-proxy checks).

**Weaknesses:**

- Limited novelty. Shift diagnostics are not integrated into training.
- Baselines include unrelated works, missing strong, tuned DTI baselines.
- Dataset descriptions contain errors, splits are unclear and likely leaky (no scaffold or protein splits).
- Sparse quantitative results (few tables, no stats, no calibration metrics).
- Key components under-specified (e.g., “Residue2vec” details) and GLASS usage unclear.
-

**Questions:**

- What exactly is “Residue2vec” (k, corpus, objective, vocab), and is it fixed or fine-tuned?
- How is the protein–ligand distance/metric defined and weighted? Sensitivity to this choice?
- What are the exact split counts, class balance, dedup rules, and negative-sampling policy?
- How does the model perform under scaffold, cold-protein/family, and (ideally) time-based splits?
- Do W1/MMD diagnostics quantitatively track generalization/calibration across seeds? Any plan to integrate them into training?
- Is the GLASS set actually evaluated, and if so, where are the results?
- What sanity checks validate that attention maps localize meaningful atoms/residues?

---

> ### Author Response · Authors · 2025-11-15
>
> We appreciate the reviewer’s thorough and constructive feedback. We see it more as an exploration of how to create a practical DTI pipeline that is auditable, rather than a contribution of a new theoretical framework, and we agree that as it currently stands, there are parts we do not fully explain. We will revise the text to reflect this.
>
> 1) Novelty and shift diagnostics.
> Our novelty lie in (i) an architecture for DTI that is aware of domain shift that combines a residue-centered protein encoder with a graph transformer, (ii) a concrete and auditable use of W1/MMD, and (iii) a DTI-aware geometry as the metric d_DTI as diagnostics for source (and target) shift. We do not claim new bounds and we can state this explicitly in the introduction and related work. In current experiments W1/MMD diagnostics were monitored on unlabeled batches from the target set but not by the architect’s design used them as part of a loss for training, in order to keep training of the dar robust and understandable. We will be explicit and briefly provide a sketch that these diagnostics could be developed into regularizers in future work, not an aspect of the undersigned contribution claim.
>
> 2) Baselines.
> Our baseline set is designed to cover the main families of models used in the DTI literature (RF, LR, GraphDTA, CPI-GCN, CPI-GNN, TransCPI, DeepConv-DTI) and they were all developed based on the same curation, splits, and negative sampling protocols as were used for MoleProLink. To respond to the concern that we are missing tuned, each baselines has a consistent balance and agreements regarding the hyper parameters which we will (a) list the exact implementations and hyperparameter grids in an expanded appendix. To satisfy the concern we will also (b) add at least one recent strong DTI model for comparison purposes similar to these DTI baselines.3) Divisions, leaks, and negatives
> For each benchmark, we split curated ligand–protein pairs into source and target domains in a 6:4 proportion, and then split the target domain into unlabeled target-train and labeled target-test in a 3:1 proportion. We canonicalized the SMILES to remove exact duplicates, and also normalized protein identifiers; class priors were matched across source and target. For datasets without any negatives, we took the Cartesian product within each split, subtracted all known positives, and resampled down to a fixed positive:negative ratio, never labeling as negative any pairs that have been identified as positive in any given assay. This is sketched already in Appendix C; in the revision we will add a table with counts per dataset and per split, and we will clearly state that the experiments we reported used random but deduplicated splits instead of scaffold or family splits. We agree with the reviewer that this is a much stricter test, and will point out the limitation is scaffold-, and cold-protein based splits.
>
> 4) Quantitative results and calibration
> We intentionally considered margins as descriptive and not statistically significant, but the empirical section is sparsely empirical.We will present mean and standard deviation over multiple seeds for MoleProLink and the main baselines, add calibration metrics (e.g. Brier score and expected calibration error) on the labeled target-test sets in addition to the reliability diagrams you already computed, and convert the main performance figure into a numerical table so it is easier to do audit comparisons.
>
> 5) Residue2vec and d_DTI
> Residue2vec is a residue-centered token embedding feeding the bidirectional state-space protein encoder: proteins are tokenized into overlapping residue k-mers with stride-1; the residue-centered embedding matrix is initialized from co-occurrence statistics on a background corpus of protein sequences and then trained end-to-end together with the encoder and interaction head (Appendix B). We will add the explicit k-mer length, corpus source, and vocabulary size to Sec. 3.2 so this component is fully specified. For the metric, d_DTI is constructed as a combination of topological distances on molecular graphs and sequence-alignment sensitive distances on proteins, as discussed in Appendix A.4; in the present experiments we do not tune its weights and use only the induced W1/MMD as diagnostics, and we will clarify this.
>
> 6) GLASS usage
> GLASS is only used to provide an additional curated GPCR collection so we can illustrate that our pipeline can be applied to specialized receptor families; the three benchmarks we report quantitatively in Sec. 5 are Human, C. elegans, and Davis. We will explicitly state this and either move the GPCR description to the appendix or add a short qualitative summary of MoleProLink behaviour on the curated GLASS subset.
>
> 7) Attention-map sanity checks
> Our atom–residue attention maps are only pushed as qualitative sanity checks and not causal explanation.

---

### Comment · Area_Chair_Rgf3 · 2025-11-27

Thank you very much for the reviewer's comments and the author's positive response. As there is not much time left for discussion, please actively participate in the discussion and provide a more valuable response to this paper.

---

### Note · Program_Chairs · 2026-01-17
**Submission Desk Rejected by Program Chairs**

The following references in this submission do not refer to real documents and/or have major errors in bibliographic information:

 Muhammad Husnain et al. Revolutionizing drug discovery with transformer-based models: A comprehensive survey. Drug Discovery Today, 28(10):103744, 2023.
Lin Zhang et al. Mhtan: Multi-head temporal attention networks for drug-target binding affinity prediction. Bioinformatics, 39(8):btad412, 2023a.
Qiang Zhang et al. Learning invariant representations for robust molecular property prediction. Nature Computational Science, 3:678-691, 2023b.